# A Laser Plane Attitude Evaluation Method for Rail Profile Measurement Sensors

**DOI:** 10.3390/s23104586

**Published:** 2023-05-09

**Authors:** Le Wang, Hao Wang, Qiang Han, Yue Fang, Shengchun Wang, Ning Wang, Guoqing Li, Shengwei Ren

**Affiliations:** Infrastructure Inspection Research Institute, China Academy of Railway Sciences Corporation Limited, Beijing 100081, China; 15121630@bjtu.edu.cn (L.W.);

**Keywords:** laser plane, attitude evaluation, coplanarity evaluation, rail profile

## Abstract

The non-coplanar lasers on both sides of the rail during full-section rail profile measurement based on line-structured light vision will cause the measured profile to be distorted, resulting in measurement errors. Currently, in the field of rail profile measurement, there are no effective methods for evaluating laser plane attitude, and it is impossible to determine the degree of laser coplanarity quantitatively and accurately. This study proposes an evaluation method based on fitting planes in response to this problem. Real-time fitting of laser planes with three planar targets of different heights provides information about the laser plane attitude on both sides of the rails. On this basis, laser coplanarity evaluation criteria were developed to determine whether the laser planes on both sides of the rails are coplanar. Using the method in this study, the laser plane attitude can be quantified and accurately assessed on both sides, effectively resolving the problem with traditional methods that can only assess the laser plane attitude qualitatively and roughly, thereby providing a solid foundation for calibration and error correction of the measurement system.

## 1. Introduction

Rails play an important role in the maintenance and repair of railway lines. As a result of regularly inspecting the rail profile, as well as evaluating the state parameters of these rails, such as vertical wear and side wear [1,2,3], it is possible to gain a better understanding of not only the state of the rails but also how to grind the rails, as this is a crucial part of railway operations and maintenance [4,5]. The measurement of rail profiles using line-structured light vision is based on the principle of triangulation and features high speed, high precision, and noncontact. As a mainstream method of dynamic detection of rail profiles globally, it can detect the parameters of in-service rails, such as vertical wear and side wear [6,7,8]. In most cases, a line-structured light sensor is placed on each side of a rail in order to obtain the profile data of the left and right half-sections of the rail. The full-section profile of the rail is then produced by splicing these two half-sections together [9,10,11]. In this process, if the laser beams of the line-structured light sensors are not on the same plane on either side of the rail, the measurement profile will be distorted to a certain degree, resulting in errors in rail profiling. It is therefore necessary to accurately assess the laser plane attitude on both sides of the rail to ensure that the laser planes on both sides are coplanar in order to obtain high-precision full-section profile data. When the lasers on both sides are installed, they are incident on the calibration plate, forming a line of intersection between the two lasers. The calibration plate is marked with a scale line. In order to determine whether the laser planes are coplanar as required, it is necessary to visually observe the degree of coincidence between the intersection lines of the laser planes on both sides and the scale line of the calibration plate. Obviously, this method is limited to a qualitative analysis and cannot provide a quantitative or accurate assessment of the plane attitude of the lasers on both sides.

Currently, little attention is being paid to the attitude of the laser planes on both sides when measuring rail profiles. In most existing studies, the focus is on adjusting the lasers in order to make them coplanar. As an example, Zhan et al. proposed a mechanism and method for the adjustment of laser planes. A mechanical adjustment mechanism could be used to translate and rotate the line-structured laser [12]. Through a high-precision manual translation stage, Chen et al. were able to scan the profiles of large workpieces [13]. In addition, some scholars have been able to align the laser planes on both sides by examining calibration techniques. In one study, Wu et al. were able to calibrate profile measurement components globally through the use of special mechanical parts, such as rotating arms [14]. In their study, Zhang et al. documented the positions of the laser profiler and markers by photographing them and converting the adjustment of coplanarity into the position adjustment of the laser profiler [15]. Ju et al. calibrated the laser planes by using the contour line calibration method [16]. Wang et al. also identified a method for correcting the error in rail profile measurement caused by non-coplanar lasers. Using projection transformation, a laser non-coplanarity correction model was proposed, as well as a reference coordinate system based on the longitudinal direction of rails. Half-section profile data were projected onto an auxiliary plane perpendicular to the longitudinal direction of the rail, and the projection profile was used to correct the measurement results [17]. Currently, few reports have been published on the use of the laser plane attitude evaluation method for both sides of the rail during the full-section measurement of the rail profile. In summary, existing methods cannot quantitatively and accurately evaluate the attitude of laser planes on both sides.

The author previously studied the distribution characteristics of laser non-coplanar error in rail profile measurement sensors and proposed a correction method for laser non-coplanar error and a calibration method for rail longitudinal parameters. However, during the installation process of rail profile measurement devices, the laser coplanar adjustment operation is mainly guided by observing the degree of collinearity of the laser line with the naked eye, which has a certain degree of blindness. In order to avoid this blindness and guide laser coplanar adjustment operations, this paper proposes a quantitative evaluation method for laser plane posture. When the measuring device is installed, three planar targets of different heights are used to obtain the attitude information of both laser planes in real time. Such information is used to establish a laser coplanarity evaluation criterion, which will be used to guide the alignment of the two line-structured lasers. As opposed to naked-eye evaluations with low precision, poor real-time performance, and subjectivity, the proposed method uses computer vision evaluations for laser coplanarity adjustment, with high precision, excellent real-time performance, and visualization, which can reduce calibration errors. In light of this, it is of great importance to improve the accuracy of the measurement of rail profiles in full sections.

## 2. Basic Principle

Figure 1 presents a schematic diagram of the full-section rail profile measurement based on line-structured light vision. Both sides of the rail are equipped with a line-structured light sensor, and the lasers of both sensors are positioned in the same plane in order to obtain the left and right half-sections of the rail. In order to obtain the full-section profile of the rail, the half-section profiles on both sides are spliced together according to the calibration parameters [18]. Using the scanning motion, the rail profile for the entire railway line can be measured in the full section. The degree of coplanarity of the laser planes on both sides of the rail is an important factor in determining the accuracy of a full-section rail profile measurement system based on line-structured light vision. Ideally, the laser planes on both sides should be coplanar in order to ensure the accuracy of rail profile measurements. As shown in Figure 2, the origin is the center of the top surface of the rail, the laser plane is the XOZ plane, and the horizontal direction is the X axis of the world coordinate system, O-XYZ. The non-coplanarity of the laser planes on both sides of the rail can therefore be expressed as the rotation of the laser plane around the X and Z axes, respectively. In Figure 3, the lasers on both sides are not coplanar, in which the left laser plane is perpendicular to the longitudinal direction of the rail, but the right laser plane rotates around the Z axis, at an angle. In this case, the right laser plane does not remain perpendicular to the longitudinal direction of the rail, nor is it coplanar to the left laser plane. While the left camera still captures the profile data of the cross-section perpendicular to the rail longitudinal direction, the right camera captures the profile data not perpendicular to the rail longitudinal direction, referred to as the oblique-section profile data. In comparison with the cross-section profile of the rail, the oblique-section profile is stretched in a certain direction, and this stretching direction is directly related to the angle between the laser planes and the longitudinal direction of the rail. As a result of this stretching, the rail profile is distorted, causing deviations in the positioning of the feature points, leading to an increase in the measurement error of the rail wear. More generally, the measured profiles on both sides of the rail are not the cross-section profile of the rail when the laser planes are not coplanar or perpendicular to the longitudinal direction of the rail. In addition, the measurements will be distorted, resulting in greater errors in determining rail profile.

At present, the laser coplanarity of the rail profile measurement system is determined by observing the degree of overlap of two laser lines with the human eye. As shown in Figure 4, the surface of the aluminum alloy ruler is engraved with a long scale line. When installing the laser scan sensors on both sides, the laser planes on both sides are projected onto the same scale line of the aluminum alloy ruler, as shown in Figure 5. The intersection line of the laser planes on both sides and the scale line of the aluminum alloy ruler are observed with the naked eye to determine whether the laser planes on both sides meet the coplanar installation requirements based on the degree of overlap between the intersection line and the scale line. Obviously, this method evaluates the attitude of a two-dimensional plane through one-dimensional lines, which can only be qualitatively evaluated and cannot be quantitatively evaluated and has a certain degree of blindness. Due to the low level of visualization, it cannot effectively guide the coplanar installation operation of two laser scan sensors. The main purpose of this article is to propose a quantitative evaluation method for laser plane attitude, which visualizes and quantifies the adjustment process of the laser plane, thereby guiding the coplanar installation operation of two laser scan sensors.

## 3. Materials and Methods

As can be seen from the analysis above, the existing method is based on visual observation of whether the lasers on both sides are coplanar; however, there are many uncertainties involved in the evaluation process, which cannot guarantee the accuracy of calibration. Therefore, this study proposes a method for evaluating laser coplanarity based on fitting planes, which can serve as a guide for the adjustment of laser coplanarity on both sides. Figure 6 illustrates the visualized laser plane adjustment device for the full-section rail profile measurement system. It contains an additional target compared to the existing full-section rail profile measurement device. The special target consists of a convex calibration block and three planar calibration plates. There are three upper surfaces on the convex calibration block, and the upper surface in the center is higher than the upper surfaces on both sides. As shown in Figure 7, three calibration plates are placed on the three upper surfaces, which are numbered 1, 2, and 3, from left to right. Consequently, the corresponding target coordinate systems, *tcs*1, *tcs*2, and *tcs*3, are established, with the center of each calibration plate as the origin and the target plane as the XOY plane.

An overview of the laser plane attitude evaluation for the full-section rail profile measurement system is presented in Figure 8. This system consists of a system calibration module, an image acquisition module, and a coplanarity evaluation module. The specific realization process for each module is described in more detail below.

The system calibration module is responsible for obtaining the internal and external parameters of the cameras. The calibration method previously described [19] is employed in this study to simultaneously collect images of the planar target calibration plates in different attitudes through the left and right cameras in order to obtain the cameras’ internal parameters.

The convex calibration block is located in the common field of view of the left and right cameras, ensuring that the left laser plane intersects the target planes of *tcs*1 and *tcs*2, and the right laser plane intersects the target planes of *tcs*2 and *tcs*3. After the lasers on both sides are turned off, the left and right cameras are used to capture images of the convex calibration block. Due to occlusion, as shown in Figure 9, the left camera can capture the entire calibration plates No. 1 and No. 2, but only a portion of calibration plate No. 3, and the right camera can capture the entire calibration plates No. 2 and No. 3, but only a portion of calibration plate No. 1. The coordinate systems of the left and right cameras are expressed as *ccs*1 and *ccs*2, respectively, and $R_{tcs1}^{ccs1},t_{tcs1}^{ccs1}$ represent the rotation matrix and translation vector of the coordinate systems *ccs*1 and *tcs*1. Since the internal parameters of the cameras are known, based on the camera calibration method as previously described, the rotation matrix, $R_{tcs1}^{ccs1},R_{tcs2}^{ccs1}$, and translation vector, $t_{tcs1}^{ccs1},t_{tcs2}^{ccs1}$, of the coordinate systems from the left camera *ccs*1 to No. 1 and No. 2 targets, as well as the rotation matrix, $R_{tcs2}^{ccs2},R_{tcs3}^{ccs2}$, and translation vector, $t_{tcs2}^{ccs2},t_{tcs3}^{ccs2}$, of the coordinate systems from the right camera *ccs*2 to No. 2 and No. 3 targets are calculated based on the images of the convex calibration block.

The image acquisition module is used to acquire the real-time light stripe images of the convex calibration block during the process of adjusting the laser planes. During the calibration process, the positions of the convex calibration block, the planar calibration plates, and the cameras remain unchanged, while the lasers on both sides are turned on and adjusted as required. With a suitable exposure time, the cameras on both sides are used to collect the light strip images of the convex calibration block in real time. The light strip image sequence of the convex calibration block is denoted as follows:(1)$$I=\left\{I_{ij}/i=1,2,j=1,2,3\ldots n\right\}$$
where i=1 is the light stripe image of the convex calibration block collected by the left camera, i=2 is the light stripe image of the convex calibration block collected by the right camera, and n is the number of their respective light stripe images. The intersection line of the laser planes and the calibration plates form a light stripe, as shown in Figure 10. The light stripe image of the convex calibration block collected by the left camera shows the intersection lines between the left laser plane and the planar calibration plates No. 1 and No. 2, which are designated as *l*1 and *l*2, respectively. The light stripe image of the convex calibration block collected by the right camera shows the intersection lines between the right laser plane and the planar calibration plates No. 2 and No. 3, which are denoted as *r*2 and *r*3, respectively.

Based on the system calibration parameters and the convex calibration block light stripe image, the coplanarity evaluation module calculates the parameters of the left and right laser planes in order to determine whether the lasers on both sides are coplanar. The coplanarity evaluation module operates in six steps, according to the data processing flow.

Step 1: The pixel coordinates of the light stripe centers are obtained by extracting the centers of the left and right light stripe images of the convex calibration block. In Figure 10, the light strip images of the convex calibration block collected by the cameras on both sides are shown, and they indicate the intersection lines *l1, l2*, *r*2, and *r*3. As follows, the light stripe centers are extracted using traditional algorithms (such as maximum value methods, grayscale center-of-gravity methods, Steger methods, template-matching methods, etc.), and the pixel coordinates of the left and right light stripe centers are obtained.
(2)$$P_{i}=\left\{\left.{\left(u_{i},v_{i}\right)}^{T}\left|0\leq u\leq width-1,0\leq v\leq height-1\right.,i=1,2\right\}\right.$$
where the width is the image width, and the height is the image height; $P_{1}$ is any point on the center of the left light stripe image; and $P_{2}$ is any point on the center of the right light stripe image.

Step 2: The pixel coordinates of the left and right light stripe image centers of the convex calibration block are converted into the corresponding target coordinate system. Different calibration plates correspond to different external parameters. The coordinate transformation process is described below, using the light strip image of the convex calibration block captured by the left camera in Figure 10 as an example. First, as shown in Figure 10, the light stripe *l*1 of the calibration plate No. 1 and the light stripe *l*2 of the calibration plate No. 2 are located. Then, for the pixel coordinate of the light stripe center of the calibration plate No. 1, ${\left(u_{i},v_{i}\right)}^{T}$, the pixel coordinate of the center of the light stripe *l*1, ${\left(u_{i},v_{i}\right)}^{T}$, is transformed into the coordinate system of the No. 1 target according to the internal and the external parameters ($R_{tcs1}^{ccs1}$ and $t_{tcs1}^{ccs1}$) of the camera. Similarly, the pixel coordinate of the center of the light stripe *l*2, $\left(u_{i},v_{i}\right)$, is transformed into the coordinate system of the No. 2 target according to the internal and external parameters ($R_{tcs2}^{ccs1}$ and $t_{tcs2}^{ccs1}$) of the camera. A similar coordinate transformation is applied to the light stripe image of the convex calibration block captured by the right camera. As a result, we have the coordinate of the intersection line *l*1 in the coordinate system of target No. 1, *tcs*1 ($P_{tcs1}^{l1}={\left(x_{tcs1}^{l1},y_{tcs1}^{l1},z_{tcs1}^{l1}\right)}^{T}$); the coordinate of the intersection line *l*2 in the coordinate system of target No. 2, *tcs*2 ($P_{tcs2}^{l2}={\left(x_{tcs2}^{l2},y_{tcs2}^{l2},z_{tcs2}^{l2}\right)}^{T}$); the coordinate of the intersection line *r*2 in the coordinate system of target No. 2, *tcs*2 ($P_{tcs2}^{r2}={\left(x_{tcs2}^{r2},y_{tcs2}^{r2},z_{tcs2}^{r2}\right)}^{T}$); and the coordinate of the intersection line *r*3 in the coordinate system of target No. 3, *tcs*3 ($P_{tcs3}^{r3}={\left(x_{tcs3}^{r3},y_{tcs3}^{r3},z_{tcs3}^{r3}\right)}^{T}$).

Step 3: The coordinate system of target No. 2 is regarded as the world coordinate system, *wcs*, and the coordinates of the light stripe centers in the respective target coordinate systems are transformed into the world coordinate system. The transformation relationship between the coordinate systems *tcs*2, *tcs*1, and *tcs*3 is calculated. The rotation matrix, $R_{tcs1}^{tcs2}$, and translation vector, $t_{tcs1}^{ccs2}$, from the target coordinate system *tcs*2 to the target coordinate system *tcs*1 are calculated by Equation (3):(3)$$\left\{\begin{array}{l} R_{tcs1}^{tcs2}={\left(R_{tcs2}^{ccs1}\right)}^{-1}\cdot R_{tcs1}^{ccs1}\\ t_{tcs1}^{tcs2}={\left(R_{tcs2}^{ccs1}\right)}^{-1}\cdot \left(t_{tcs1}^{ccs1}-t_{tcs2}^{ccs1}\right) \end{array}\right.$$

The rotation matrix and translation vector from the target coordinate system *tcs*2 to the target coordinate system *tcs*3 are calculated by Equation (4):(4)$$\left\{\begin{array}{l} R_{tcs3}^{tcs2}={\left(R_{tcs2}^{ccs2}\right)}^{-1}\cdot R_{tcs3}^{ccs2}\\ t_{tcs3}^{tcs2}={\left(R_{tcs2}^{ccs2}\right)}^{-1}\cdot \left(t_{tcs3}^{ccs2}-t_{tcs2}^{ccs2}\right) \end{array}\right.$$

The world coordinates of the light stripe center on the intersection lines *l*2 and *r*2 are the same as the coordinates in the target coordinate system, that is, $P_{wcs}^{l2}=P_{tcs2}^{l2}$ and $P_{wcs}^{r2}=P_{tcs2}^{r2}$. The coordinates of the intersection lines *l*1 and *r*3 in the world coordinate system, *wcs*, namely $P_{wcs}^{l1}={\left(x_{wcs}^{l1},y_{wcs}^{l1},z_{wcs}^{l1}\right)}^{T}$ and $P_{wcs}^{r3}={\left(x_{wcs}^{r3},y_{wcs}^{r3},z_{wcs}^{r3}\right)}^{T}$, are obtained by Equations (5) and (6), respectively. Similarly, we have the coordinate of the intersection line *l*2 in the world coordinate system ($P_{wcs}^{l2}={\left(x_{wcs}^{l2},y_{wcs}^{l2},z_{wcs}^{l2}\right)}^{T}$), the coordinate of the intersection line *r*2 in the world coordinate system ($P_{wcs}^{r2}={\left(x_{wcs}^{r2},y_{wcs}^{r2},z_{wcs}^{r2}\right)}^{T}$), and the coordinate of the intersection line *r*3 in the world coordinate system ($P_{wcs}^{r3}={\left(x_{wcs}^{r3},y_{wcs}^{r3},z_{wcs}^{r3}\right)}^{T}$).
(5)$$\left[\begin{array}{c} P_{wcs}^{l1}\\ 1 \end{array}\right]=\left[\begin{array}{c} x_{wcs}^{l1}\\ y_{wcs}^{l1}\\ z_{wcs}^{l1}\\ 1 \end{array}\right]=\left[\begin{array}{cc} R_{tcs1}^{tcs2} & t_{tcs1}^{tcs2}\\ \begin{array}{ccc} 0 & 0 & 0 \end{array} & 1 \end{array}\right]\left[\begin{array}{c} P_{tcs1}^{l1}\\ 1 \end{array}\right]=\left[\begin{array}{cc} R_{tcs1}^{tcs2} & t_{tcs1}^{tcs2}\\ \begin{array}{ccc} 0 & 0 & 0 \end{array} & 1 \end{array}\right]\left[\begin{array}{c} x_{tcs1}^{l1}\\ y_{tcs1}^{l1}\\ z_{tcs1}^{l1}\\ 1 \end{array}\right]$$
(6)$$\left[\begin{array}{c} P_{wcs}^{r3}\\ 1 \end{array}\right]=\left[\begin{array}{c} x_{wcs}^{r3}\\ y_{wcs}^{r3}\\ z_{wcs}^{r3}\\ 1 \end{array}\right]=\left[\begin{array}{cc} R_{tcs3}^{tcs2} & t_{tcs3}^{tcs2}\\ \begin{array}{ccc} 0 & 0 & 0 \end{array} & 1 \end{array}\right]\left[\begin{array}{c} P_{tcs3}^{r3}\\ 1 \end{array}\right]=\left[\begin{array}{cc} R_{tcs3}^{tcs2} & t_{tcs3}^{tcs2}\\ \begin{array}{ccc} 0 & 0 & 0 \end{array} & 1 \end{array}\right]\left[\begin{array}{c} x_{tcs3}^{r3}\\ y_{tcs3}^{r3}\\ z_{tcs3}^{r3}\\ 1 \end{array}\right]$$

Step 4: In the world coordinate system, the two intersection lines (*l*1 and *l*2) of the left laser plane and the two intersection lines (*r*2 and *r*3) of the right laser plane are sequentially fitted to the plane, and the parameters of the left and right laser planes are obtained. The fitting process is described below, taking the intersection lines *l*1 and *l*2 as an example.

In $P_{wcs}^{i}={\left(x_{wcs}^{i},y_{wcs}^{i},z_{wcs}^{i}\right)}^{T},i=1,2,3\ldots k$ is any point on the intersection lines *l*1 and *l*2 of the left laser plane and the calibration plates No. 1 and No. 2, where k=m+n, and *m* and *n* are the number of points on the intersection lines *l*1 and *l*2, respectively. The following matrix is thus constructed:(7)$$M=\left[\begin{array}{c} x_{wcs}^{1}-\bar{x},x_{wcs}^{1}-\bar{y},x_{wcs}^{1}-\bar{z}\\ \vdots \\ x_{wcs}^{k}-\bar{x},y_{wcs}^{k}-\bar{y},x_{wcs}^{1}-\bar{z} \end{array}\right]$$
where $\bar{x}=\frac{1}{k}\sum_{i=1}^{k}x_{wcs}^{i},\bar{y}=\frac{1}{k}\sum_{i=1}^{k}y_{wcs}^{i},\bar{z}=\frac{1}{k}\sum_{i=1}^{k}z_{wcs}^{i}$. The point $\bar{y}{\left(\bar{x},\bar{y},\bar{z}\right)}^{T}$ is designated as the center of gravity of the plane. If $S=M^{T}\cdot M$, where *S* has three eigenvalues, then the eigenvector corresponding to the smallest eigenvalue is the normal of the fitting plane, $n^{l}$. The left laser plane is constructed with ${\left(\bar{x},\bar{y},\bar{z}\right)}^{T}$ as a point on the plane and the vector $n^{l}$ as the normal. Similarly, the intersection lines *r*2 and *r*3 of the right laser plane are fitted to the plane to obtain the normal of the right laser plane, $n^{r}$, and the right laser plane is constructed.

Step 5: The angle between the normals of the fitting planes on the left and right sides (α) and the distance between the planes (d) are calculated. The angle, α, can be calculated by Equation (8):(8)$$\alpha =\arccos\left(\frac{n^{l}\cdot n^{r}}{\left|n^{l}\right|\left|n^{r}\right|}\right)$$

In order to calculate the distance between the two planes, the corresponding plane coordinate systems *pcs*1 and *pcs*2 are established with the center of gravity of the fitting planes on the left and right sides as the origin and the normal direction as the Z axis. The rotation of the coordinate system around the Z axis will not affect the direction of the plane normal in the plane coordinate system. For simplicity, the rotation around the Z axis is set to 0 here. The rotation matrix and translation vector from the plane coordinate system *pcs*1 to the world coordinate system (*wcs*) are denoted as $R_{wcs}^{pcs1}$ and $t_{wcs}^{pcs1}$, and the rotation matrix and translation vector from the world coordinate system (*wcs*) to the plane coordinate system *pcs*2 are expressed as $R_{pcs2}^{wcs}$ and $t_{pcs2}^{wcs}$. As a result, the corresponding homogeneous transformation matrices are constructed, as shown below.
(9)$$H_{wcs}^{pcs1}=\left[\begin{array}{cc} R_{wcs}^{pcs1} & t_{wcs}^{pcs1}\\ \begin{array}{ccc} 0 & 0 & 0 \end{array} & 1 \end{array}\right]$$
(10)$$H_{pcs2}^{wcs}=\left[\begin{array}{cc} R_{pcs2}^{wcs} & t_{pcs2}^{wcs}\\ \begin{array}{ccc} 0 & 0 & 0 \end{array} & 1 \end{array}\right]$$

Then the homogeneous transformation matrix of the coordinate system *pcs*1 and *pcs*2 can be expressed as follows:(11)$$H_{pcs2}^{pcs1}=H_{wcs}^{pcs1}\cdot H_{pcs2}^{wcs}$$

Therefore, for any point in the coordinate system *pcs*2, $P_{pcs2}={\left(x_{pcs2},y_{pcs2},z_{pcs2}\right)}^{T}$, it can be transformed into the coordinate system *pcs*1 by Equation (12), where $P_{pcs1}={\left(x_{pcs1},y_{pcs1},z_{pcs1}\right)}^{T}$ is the corresponding coordinate of the point in the coordinate system *pcs*1.
(12)$$\left[\begin{array}{c} P_{pcs1}\\ 1 \end{array}\right]=\left[\begin{array}{c} x_{pcs1}\\ y_{pcs1}\\ z_{pcs1}\\ 1 \end{array}\right]=H_{pcs2}^{pcs1}\cdot P_{pcs2}=H_{pcs2}^{pcs1}\left[\begin{array}{c} x_{pcs2}\\ y_{pcs2}\\ z_{pcs2}\\ 1 \end{array}\right]$$

*N* points on the right laser plane are arbitrarily taken, as shown in Equation (13), and transformed into the coordinate system *pcs*1 through Equation (9). Then, we have Equation (14).
(13)$$P_{pcs2}^{i}={\left(x_{pcs2}^{i},y_{pcs2}^{i},z_{pcs2}^{i}\right)}^{T},i=1,2,3\ldots N$$
(14)$$P_{pcs1}^{i}={\left(x_{pcs1}^{i},y_{pcs1}^{i},z_{pcs1}^{i}\right)}^{T},i=1,2,3\ldots N$$

Since the fitting plane on the left coincides with the *XOY* plane of the coordinate system *pcs*1, $\left|z_{pcs1}^{i}\right|,i=1,2,3\ldots N$ is the distance from these *N* points to plane 1, and the distance (*d*) from plane 2 to plane 1 can be expressed as follows:(15)$$d=\frac{1}{N}\sum_{i=1}^{i=N}\left|z_{pcs1}^{i}\right|$$

Step 6: Both the distance between the planes and the angle between the normals determine whether the lasers on both sides are coplanar. If the angle between the two planes is 0, it means that the two planes are parallel or coincident. If the distance from any point on one of the planes to the other plane is 0, the two planes coincide. To eliminate the influence of errors, two parameters are used to determine whether the lasers on both sides are coplanar, namely the angle (α) and the distance (d). As long as the angle and distance satisfy Equation (16), the coincidence degree of the left and right laser planes is high, and the laser planes on both sides of the rail are appropriate.
(16)$$d\leq T_{d}\;\&\;\alpha \leq T_{\alpha }$$
where $T_{d},\;T_{\alpha }$ are the thresholds of the distance (d) and the angle (α), which can be determined in accordance with the accuracy requirements.

When the laser planes are adjusted, the left and right cameras collect real-time light stripe images of the convex calibration block and obtain the sequence of light stripes, as shown in Equation (2). Then, the coplanarity evaluation module processes the light strip image sequence of the convex calibration block in real time, calculates the laser plane parameters on both sides of the rail, draws the laser planes on both sides in real time in the display window, and displays the angle and distance between the normals. By utilizing Equation (16), the module determines if the laser planes on both sides of the rail are appropriate, allowing the full-section rail profile measurement system to visualize the alignment of both laser planes and effectively determine if the lasers are coplanar on both sides. The effect is shown in Figure 11.

## 4. Results and Discussion

### 4.1. Experimental Design

To confirm the advantages of the proposed laser coplanarity evaluation method, such as a strong real-time performance, high accuracy, and visualization, a laser coplanarity evaluation experiment was designed, and the experimental device shown in Figure 12 was constructed. Among them, the camera is the Ranger 3 high-speed camera produced by the German SICK company (Waldkirch, Germany), with a resolution of 2560 × 832 pixels, and the line laser is produced by the Canadian Osela company, with a wavelength of 660 nm and 450 nm, respectively. The laptop used for test is ThinkPad T480, Intel^®^, Core™, i7-8550U CPU@1.80GHz, 1.99 GHz, 32.0 GB memory, 64-bit operating system.

First, the internal parameters of the left and right cameras are calibrated based on the planar target, and then the simplified convex calibration block is placed in the common field of view of the two cameras, ensuring that the left laser plane intersects the calibration plates No. 1 and No. 2, and the right laser plane intersects the calibration plates No. 2 and No. 3. Then, the lasers on both sides are turned off, and the images of the calibration plates are collected through the cameras on both sides. The result is shown in Figure 9. After that, the lasers on both sides are turned on, and the light strip images of the three calibration plates are collected in real time through the cameras on both sides. The result is shown in Figure 10. The parameters of the laser planes on both sides are obtained in real time from the cross-sectional laser images of the three calibration plates, and the two laser planes are drawn in real time in the program window. The lasers on both sides are adjusted according to the two laser planes displayed in real time in the program window until they satisfy the coplanarity standard. The following three groups of experiments were carried out:

The first group of experiments is a real-time verification experiment. On the basis of the abovementioned experimental device, the laser planes on both sides are continuously adjusted, and two cameras collect 500 target images each in real time. At the same time, the coplanar degree of the two laser planes is calculated by the proposed method, and the real-time performance of the proposed method is evaluated by the time of program brushing the new window.

The second group of experiments is the accuracy verification experiment. In this experiment, the accuracy of the proposed method is evaluated by the laser plane fitting error, laser plane distance measurement error, and laser plane angle measurement error. First, in the process of laser plane adjustment on both sides, 500 consecutive target images are collected by two cameras, corresponding to 500 × 2 = 1000 laser plane positions; therefore, the accuracy of the proposed method is evaluated with these 1000 laser plane fitting errors. As shown in Figure 13, we kept the camera and two targets of different heights stationary and placed the line laser on a precision displacement table and a rotating table. Then, the controlled line laser moved from 0 mm translational motion to 10 mm, with a step length of 1 mm. The light stripe images of two targets were collected at each position. With the position of 0 mm as the reference, the proposed method was used to calculate the distance between the laser plane at the other 10 different positions and the laser plane at the reference position. Finally, controlling the line laser, we rotated from 0 deg to 2 deg, with a step size of 0.200 deg. Similarly, two target light stripe images were collected at each position, and the angle between the remaining ten different laser planes and the reference position laser plane was calculated using the proposed method, using the 0 deg position as a reference. In order to compare the measurement accuracy of the rail profile after laser coplanar adjustment, a comparative experiment was designed using a standard worn rail as the measurement object. The standard worn rail are shown in Figure 14, with a vertical wear of 11.00 mm. When the laser planes of two laser scan sensors are not coplanar (in order to highlight the effect, the angle between the laser planes is about 3 deg), collect the full-section profile of the steel rail 20 times and calculate its vertical wear. Under the guidance of the proposed method, two laser scan sensors were adjusted to meet the installation requirements of coplanarity. The full profile of the steel rail was collected 20 times again, and its vertical wear was calculated.

The third group of experiments is the repeatability verification experiment. Keep the measuring device stationary, continuously collect 500 pairs of target images through two cameras, and also evaluate the repeatability of the proposed method with the fitting error of the laser plane.

### 4.2. Experimental Result

#### 4.2.1. Real-Time Verification Experiment Results

Figure 15 exhibits the laser planes at three typical positions during the adjustment process. It can be seen that the program window displays the laser planes on both sides in real time and calculates the current coplanarity of the laser planes on both sides in real time, according to the standard. Therefore, as opposed to traditional methods that rely on visually observing laser beams, the proposed method allows for the visualization of the three-dimensional laser plane in real time, thus avoiding blindness caused by observing two-dimensional laser beams. Figure 16 shows the corresponding refresh window time of the two cameras. The average refresh time is 0.03 s, and the frame rate is about 33 frames/second, which can fully meet the real-time requirements.

#### 4.2.2. Accuracy Verification Experiment Results

The system calibration module calculates the transformation matrix between the camera coordinate system and the target coordinate system through a single planar target image, with known camera internal parameters. Figure 17 shows the reprojection error diagram. The average reprojection errors of the No. 1 calibration plate and the No. 2 calibration plate images obtained by the left camera are 0.037 pixel and 0.025 pixel, respectively. The average reprojection errors of the images of the No. 2 calibration plate and the No. 3 calibration plate obtained by the right camera are 0.026 pixel and 0.027 pixel, respectively, and the projection errors are within a reasonable range.

The laser plane fitting results and statistical values are given in Figure 18 and Table 1, respectively. It can be seen that, during the random change of the laser plane attitude, the average values of the laser plane fitting errors on both sides obtained by the proposed method are 0.062 mm and 0.054 mm, respectively, which are smaller than the laser plane calibration error of the rail profile measurement sensor. 

The results of laser plane distance and angle measurement are shown in Table 2, where MV represents the measured value, AV represents the actual value, and ME represents the measurement error. It can be seen that when the laser plane is moving in a translational motion, the maximum value of the distance measurement error of the laser plane is 0.13 mm, and the average error is 0.09 mm. When the laser plane is rotating, the maximum value of the angle measurement error of the laser plane is 0.019 deg, and the average value is 0.009 deg. The measurement errors of plane distance and plane angle are within an acceptable range.

The results of rail wear before and after coplanar adjustment are shown in Figure 19, and Table 3 provides the corresponding error statistics. It can be seen that, after the coplanarity adjustment, the average measurement error of the rail’s vertical wear decreased from 0.279 mm to 0.037 mm. Therefore, the proposed method can guide the laser coplanar adjustment process and improve the accuracy of rail profile measurement.

Therefore, the proposed laser coplanar evaluation method has high accuracy. Based on the accurate acquisition of laser plane parameters on both sides, the degree of laser plane on both sides can be quantitatively evaluated, which is superior to the traditional qualitative evaluation method based on visual inspection.

#### 4.2.3. Repeatability Validation Test Results

The repeatability experimental results and statistical values are shown in Figure 20 and Table 4, respectively. It can be seen that when the laser plane is not adjusted, the average values of the fitting errors of the two laser planes obtained by the proposed method are 0.022 mm and 0.021 mm, respectively, and the standard deviations are 0.013 mm and 0.012 mm, respectively, indicating that the method has high repeatability.

### 4.3. Discussions

Based on past experience, when the target area accounts for more than a quarter of the entire image, it can ensure that the target attitude evaluation has high accuracy. Conversely, when the target area accounts for a small proportion in the image, a certain degree of measurement accuracy will be lost. Due to the limitations of the on-site installation environment of the rail profile measurement sensor, larger targets cannot be used, and the target area accounts for significantly less than a quarter, which limits the accuracy of laser planar attitude assessment. Therefore, in response to the accuracy issue caused by the small proportion of target areas, the author intends to perform further research work around target optimization and attitude evaluation algorithm optimization in the next step.

Although using this method can ensure laser coplanar installation on both sides of the rail, the next question is how to ensure the laser plane perpendicular to the longitudinal direction of the rail. To address the issue, it is necessary to obtain the longitudinal direction vector of the rail in the camera coordinate system (or world coordinate system). We proposed a solution to calibrate the rail longitudinal direction in our previous research [8]. By combining the longitudinal vector parameters of the rail and the laser plane parameters, it is possible to further quantitatively evaluate whether the installation of laser scan sensors meets the requirements.

## 5. Conclusions

Considering the difficulty of evaluating laser plane attitude in the measurement of rail profiles, this paper presents a method for evaluating laser plane attitude using fitting planes, analyzes the impact of non-coplanar lasers on plane profile measurement results, elaborates on how the laser plane attitude calculation is calculated, and constructs the criterion for evaluating laser coplanarity. The experimental results indicate that the proposed method has the advantages of high accuracy, real-time performance, and excellent visualization. This method improves the laser coplanarity adjustment process from the original naked-eye evaluation with low precision, poor real-time performance and subjectivity to computer-based evaluation with high precision, and strong real-time performance and visualization. By doing so, it reduces the calibration error of the traditional calibration method and provides a theoretical basis for improving the accuracy and reliability of rail profile measurement systems. In on-site applications, factors such as changes in ambient light and target posture affect the on-site evaluation efficiency and calibration accuracy of the proposed method. Therefore, it is necessary to improve the evaluation algorithm for railway on-site application scenarios to further improve the efficiency and accuracy of on-site evaluation.

## Figures and Tables

**Figure 1 sensors-23-04586-f001:**
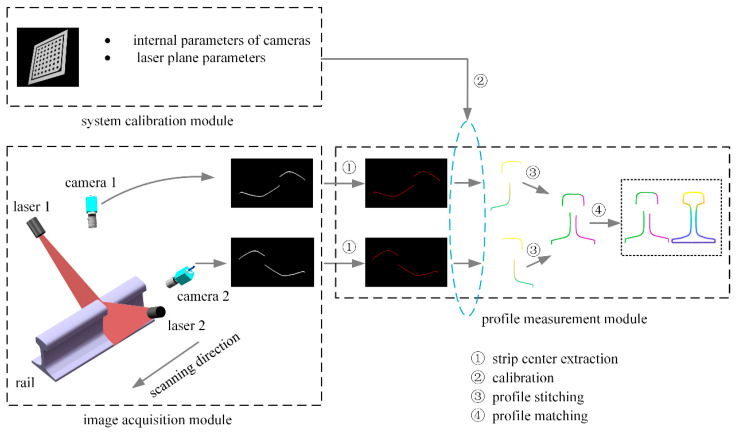
Schematic diagram of the full-section rail profile measurement based on line-structured light vision.

**Figure 2 sensors-23-04586-f002:**
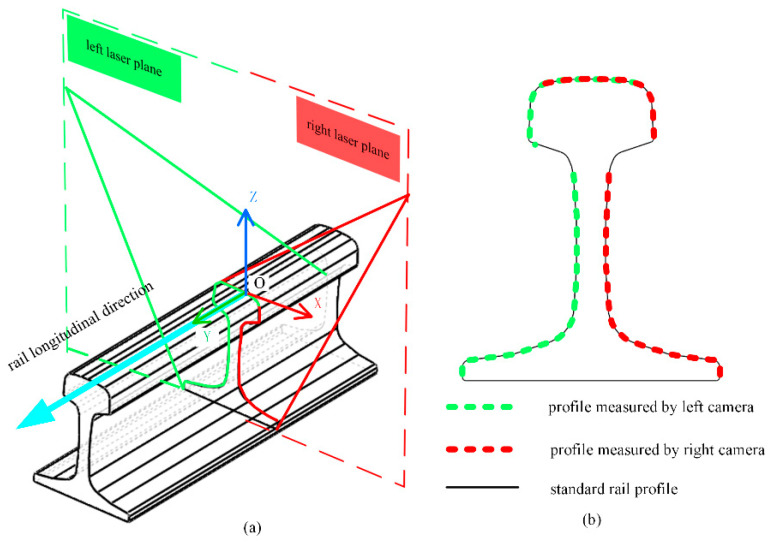
Schematic diagram of coplanar lasers on both sides: (**a**) coplanar lasers and (**b**) profile measurement results.

**Figure 3 sensors-23-04586-f003:**
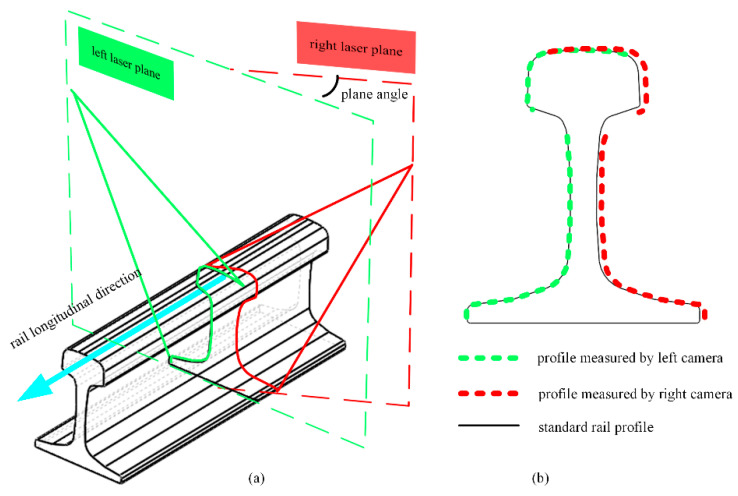
Schematic diagram of non-coplanar lasers on both sides: (**a**) non-coplanar lasers and (**b**) profile measurement results.

**Figure 4 sensors-23-04586-f004:**
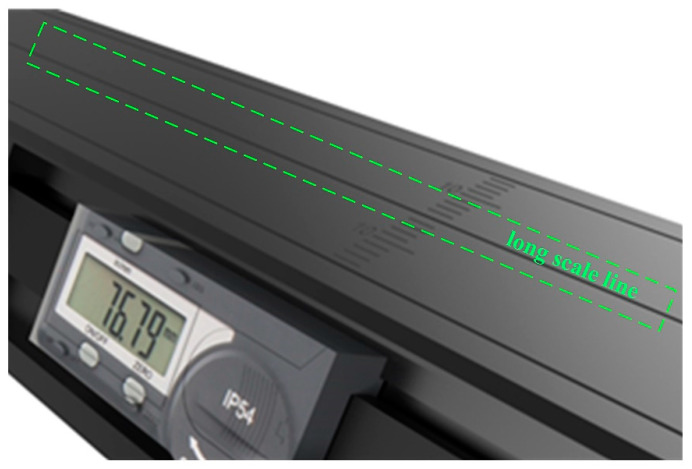
Aluminum alloy ruler.

**Figure 5 sensors-23-04586-f005:**
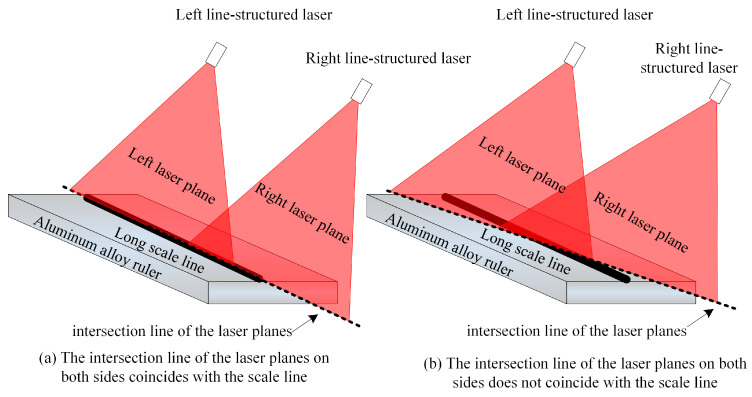
Schematic diagram of the traditional laser plane evaluation method for rail profile measurement system.

**Figure 6 sensors-23-04586-f006:**
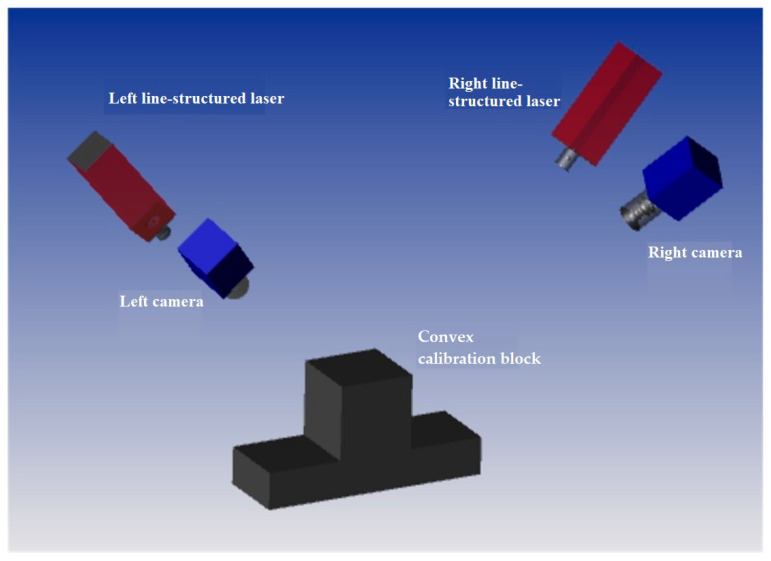
The visualized laser plane adjustment device for the full-section rail profile measurement system.

**Figure 7 sensors-23-04586-f007:**
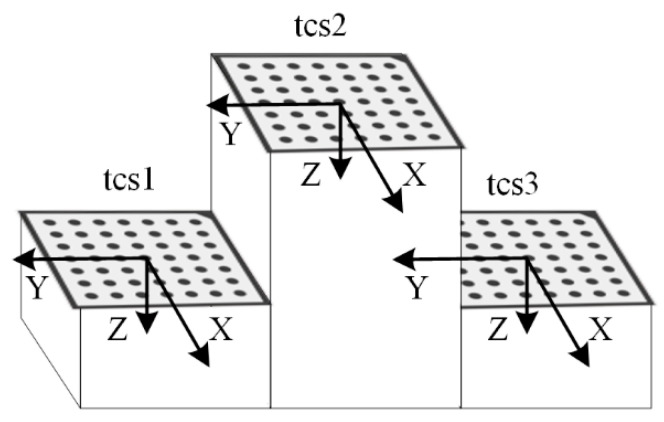
The coordinate system, which is composed of the convex calibration block and three planar targets.

**Figure 8 sensors-23-04586-f008:**
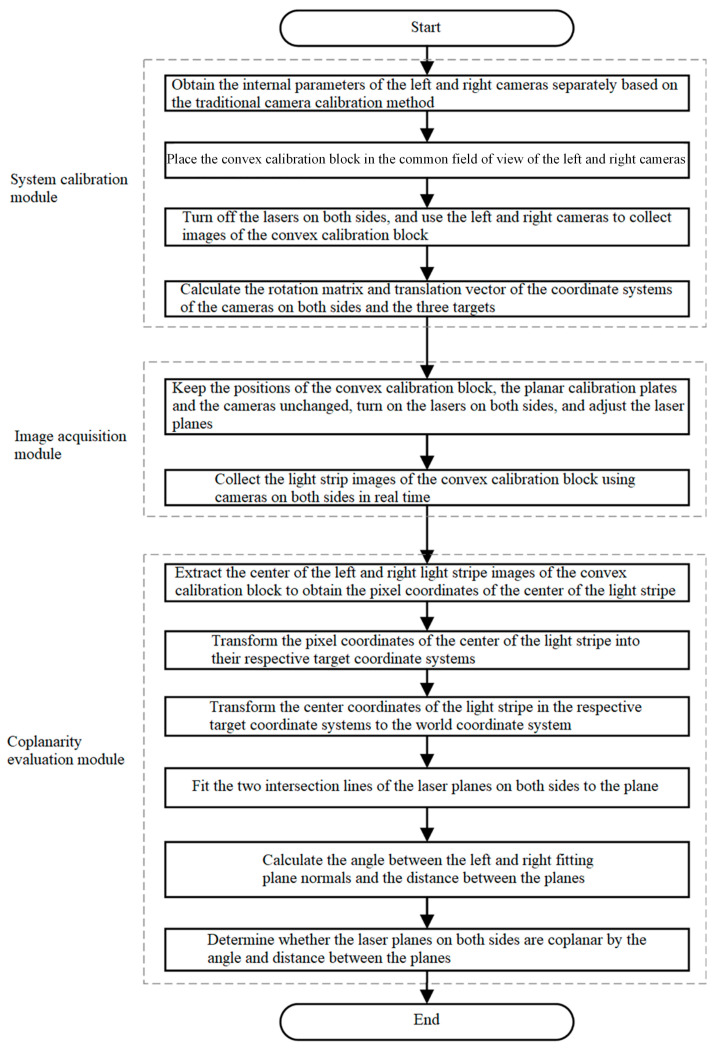
Flowchart of laser coplanarity evaluation of the full-section rail profile measurement system.

**Figure 9 sensors-23-04586-f009:**
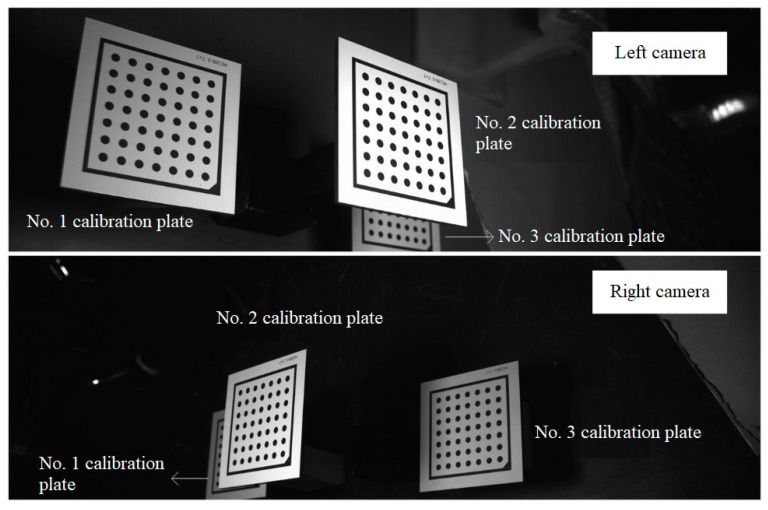
The images of the convex calibration block captured by the left and right cameras.

**Figure 10 sensors-23-04586-f010:**
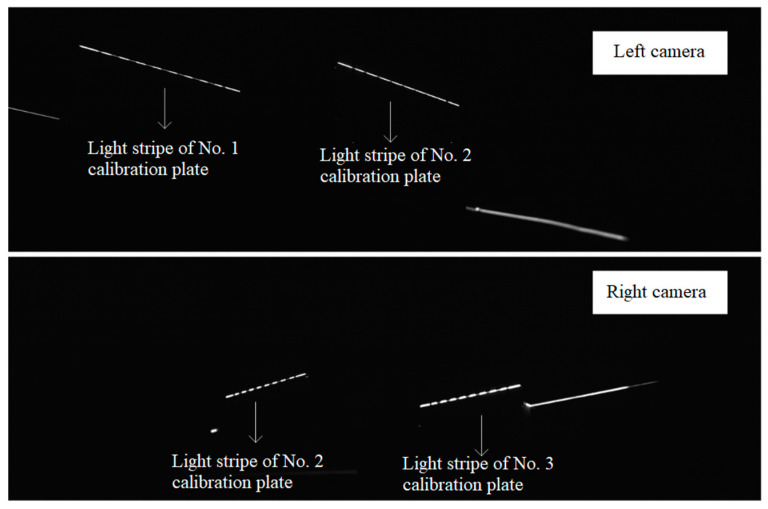
The left and right light stripe images of the convex calibration block.

**Figure 11 sensors-23-04586-f011:**
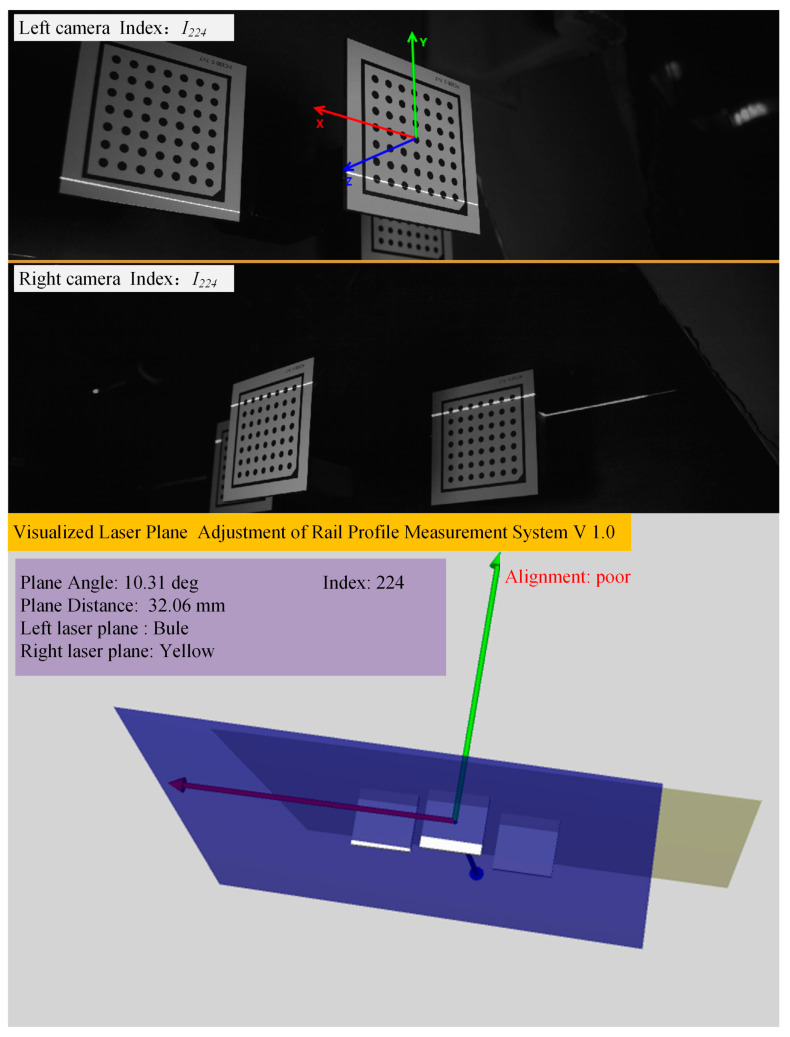
Schematic diagram of the visualized adjustment of the laser planes on both sides of the rail.

**Figure 12 sensors-23-04586-f012:**
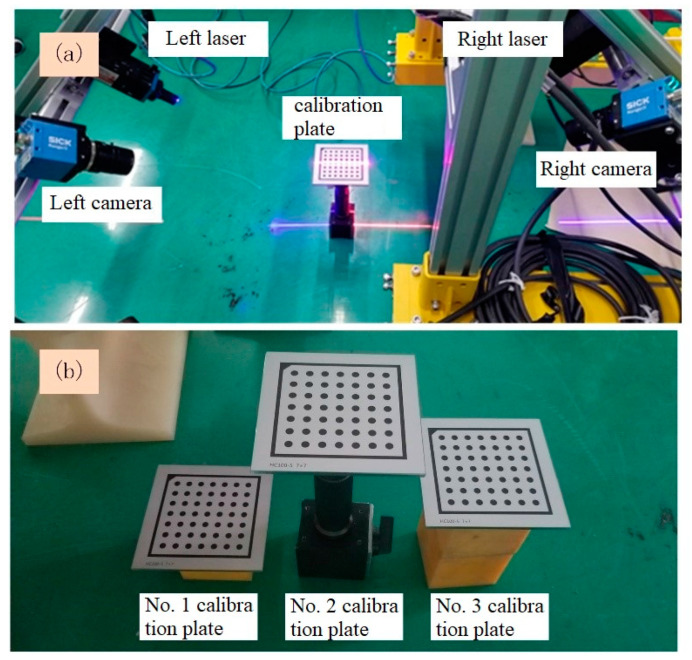
Experimental device for laser coplanarity evaluation: (**a**) calibration of the internal parameters of the cameras and (**b**) simplified convex calibration block and calibration plates.

**Figure 13 sensors-23-04586-f013:**
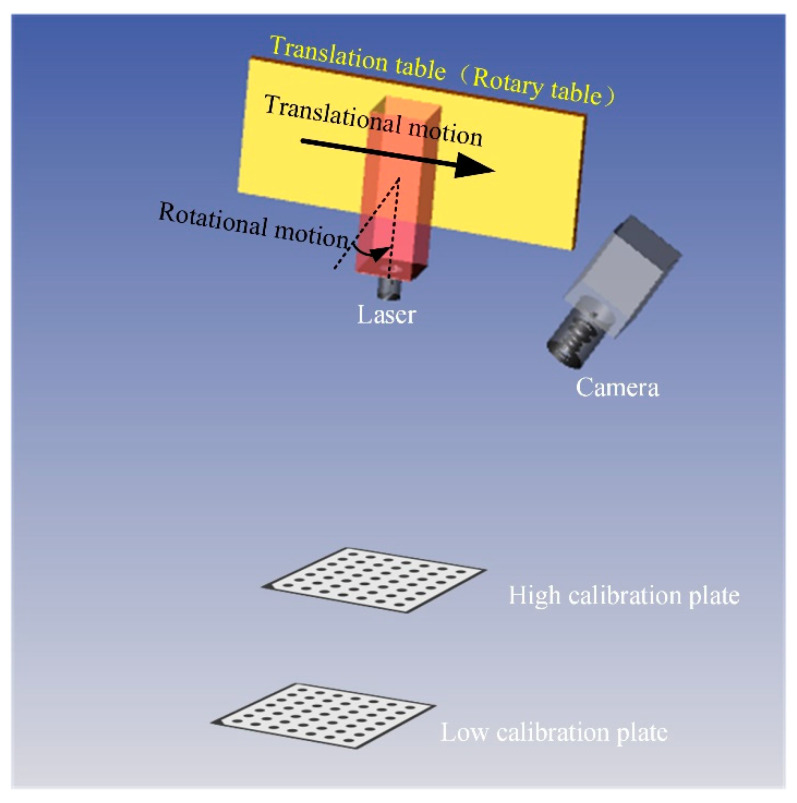
Schematic diagram of accuracy verification experiment.

**Figure 14 sensors-23-04586-f014:**
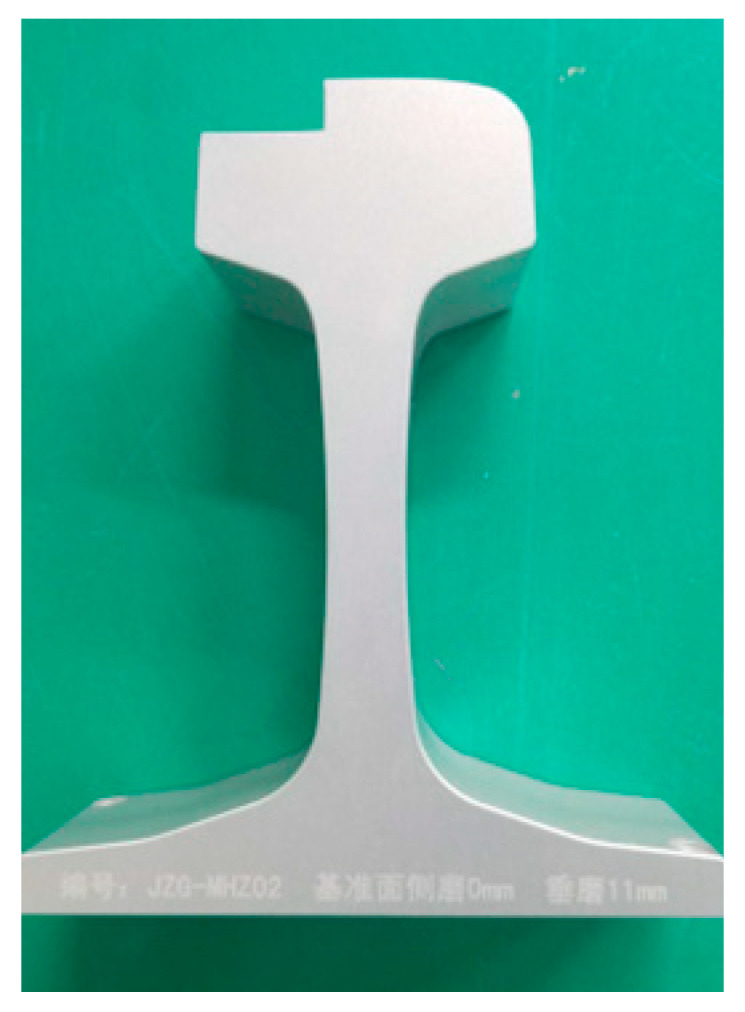
Standard worn rail with a vertical wear of 11.00 mm. (NO. JZG-MHZ02, horizontal wear 0 mm, vertical wear 11 mm).

**Figure 15 sensors-23-04586-f015:**
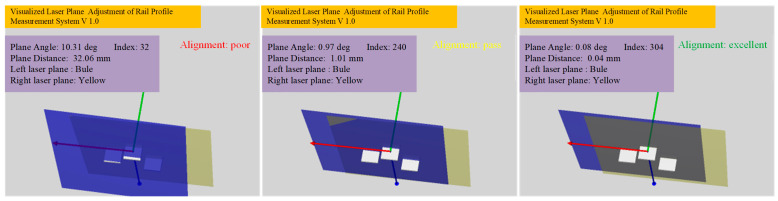
The laser planes at three typical positions.

**Figure 16 sensors-23-04586-f016:**
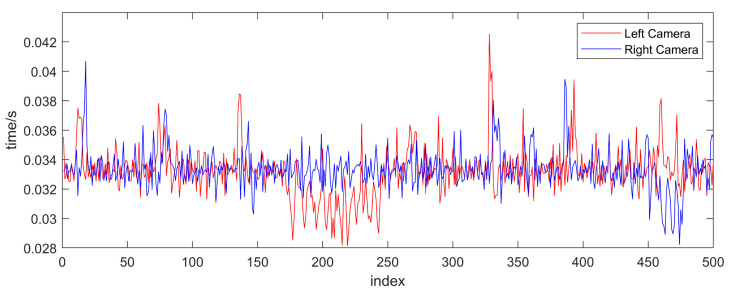
The time for the program to refresh the window.

**Figure 17 sensors-23-04586-f017:**
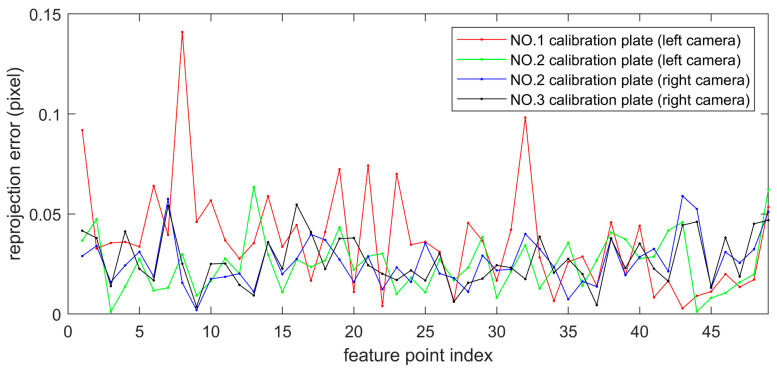
Reprojection errors of three calibration plates.

**Figure 18 sensors-23-04586-f018:**
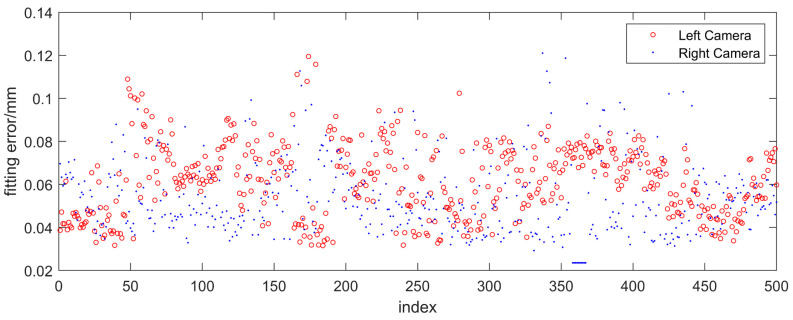
The results of laser plane fitting error.

**Figure 19 sensors-23-04586-f019:**
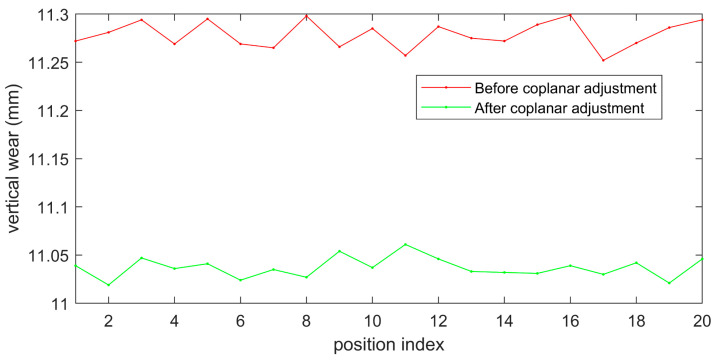
Measurement results of rail wear before and after laser coplanar adjustment.

**Figure 20 sensors-23-04586-f020:**
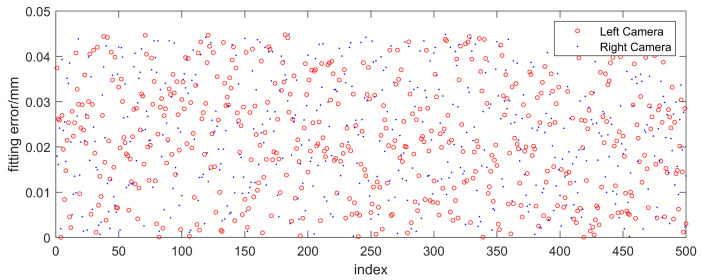
The results of the repeatability verification experiment.

**Table 1 sensors-23-04586-t001:** Statistical results of the laser plane fitting error (mm).

Camera	Average Value	Standard Deviation	Maximum
Left camera	0.062	0.017	0.119
Right camera	0.054	0.017	0.121

**Table 2 sensors-23-04586-t002:** Laser plane distance and angle measurement results.

Location Index		1	2	3	4	5	6	7	8	9	10
Distance (mm)	AV	1.00	2.00	3.00	4.00	5.00	6.00	7.00	8.00	9.00	10.00
	MV	1.11	2.09	3.06	3.87	5.06	5.94	7.14	7.93	9.12	10.06
	ME	0.11	0.09	0.06	0.13	0.06	0.06	0.14	0.07	0.12	0.06
Angle (deg)	AV	0.200	0.400	0.600	0.800	1.000	1.200	1.400	1.600	1.800	2.000
	MV	0.206	0.400	0.581	0.807	1.008	1.191	1.415	1.616	1.804	1.991
	ME	0.006	0.000	0.019	0.007	0.008	0.009	0.015	0.016	0.004	0.009

**Table 3 sensors-23-04586-t003:** Measurement results of rail wear before and after coplanar adjustment (mm).

Statistical Value	Mean Error	Standard Deviation
Before coplanar adjustment	0.279	0.014
After coplanar adjustment	0.037	0.011

**Table 4 sensors-23-04586-t004:** Statistical results of the repeatability verification experiment (mm).

Camera	Average Value	Standard Deviation	Maximum
Left camera	0.022	0.013	0.045
Right camera	0.021	0.012	0.044

## Data Availability

Data available on request from the authors.

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
