# Peer review of "A Laser Plane Attitude Evaluation Method for Rail Profile Measurement Sensors"

_sensors, 2023, doi:10.3390/s23104586_

Round 1

Reviewer 1 Report

Comments on sensors-2304433

This paper proposes a method to measure the cross-sectional profile of a rail.

Though the technologies are not innovative, It will be useful for practical measurement.

The reviewer has the following opinions.

Please reply to them.

1.     The reviewer is afraid that the laser lines are not straight. Did you check the linearity of the line..

2.     How did you analyze the accurate position of the laser light line?

Why the cross lines were used to project on the calibration block?

In this case if the line laser is only on the white line. it is not possible to calibrate.

The reviewer thinks it is better to use one-directional parallel grating instead of the cross grating.

3.     How much is the actual N value?

4.     Is Equation 16 commonly used in English? It looks Chinese character is used.

5.     How much speed is for measuring the actual rail?

If the measurement speed is 100 km/hour, the length of the rail for one frame is almost 2m for mean refresh time of 80.8ms. It is too long.

6.     The description of the unit in Figures and Tables is not appropriate.

Author Response

Thanks very much for your time to review this manuscript. I really appreciate all your comments and suggestions. We have considered these comments carefully and tried our best to address every one of them. Please see the attachment.

Reviewer 2 Report

In this article, the authors propose a method for evaluating the position of the laser plane for rail profile measurement sensors. The topic is certainly interesting for the readers of Sensors.

The structure of the manuscript is correct. It consists of the literature review the theoretical background of the method, experimental results, results discussion and the conclusions. Generally the manuscript is good.

Minor comments:

In Introduction, the author's contributions are not well described. More descriptions for the author's contributions could be added to show the differences of contributions between the previous study and the one presented in the paper.

In the conclusions authors should provide information what are their plans for the further research work. I suggest also to write a few sentences what are limitations of the method proposed by authors.

The bibliography presented is correct; however, it could be more complete.

Author Response

(The authors gave the same response as above.)

Reviewer 3 Report

This paper tries to propose a method to solve the non-coplanar problem of two laser planes in rail profile detection. However, the practical works only achieved real-time fitting of two laser planes. A lot of work should be added. And there are also some questions.

1.       The experiments of this paper are not sufficient. In abstract, the author said, “ there are no effective methods for evaluating laser plane attitude, and it is impossible to determine the degree of laser coplanarity quantitatively and accurately.” In reality, there are many methods to evaluate laser planes and therefore, it is possible to determine the degree of laser coplanarity quantitatively and accurately. But these technologies had not been used in rail profile measurement. The significance of this study should be increasing the measurement accuracy of the rail profile by fitting two laser places, not just proposing a real-time laser plane fitting method. The following verification experiments should be added.

1.1 An experiment to verify the accuracy of camera calibration. As mentioned in Line 138, the authors used Zhang’s method to calibrate the camera. The checkboard was placed in multiple locations in space to improve calibration accuracy. But in this paper, each camera only captured two calibrated plates. How about the calibration accuracy? What are the values of calibration parameters and re-project error?

1.2 In Section 4.2 Experiments (2), the authors only gave the fitting errors of each laser plane. Then the authors said, “the degree of laser plane on both sides can be quantitatively evaluated.” The laser plane fitting errors of 0.062 mm or 0.054 mm can not illustrate the attitude of the laser plane. So the experiments to show the attitude of the laser plane should be added. Furthermore, the experiments to show that the coplanar of two laser planes is modified by the guidance of proposed methods should be added.

1.3 Because this method is to increase the rail profile measurement accuracy as shown in Fig.2 and Fig.3, the experiments to show that after coplane adjustment by the proposed method, the measurement accuracy of rail profile is higher should be given.

2. In Fig.6, the second block of System calibration module is repetition.

3. In Line 209 Step 2, the pixel coordinate can not be transformed to a three-dimensional coordinate system by only one camera parameter unless the scale is lost. The author did not mention it. But Fig.9 shows “The distance between the laser planes on both sides: 32.06mm.” How was the scale restored?

4. In Eq.(2), Is P1 or P2 a point or a point set ? P1 can not express a point set.

5. In Fig.6, is the camera calibration process a repetitive process? Why did the authors have to use a convex calibration block to do camera calibration and laser stripe extraction meanwhile?

6. Does the non-parallelism of the three planes affect the proposed method?

7. In Section 2 Line 105, “More generally, the measured profiles on both sides of the rail are not the cross-section profile of the rail …”. If the two laser planes are adjusted coplanar, how to ensure the laser plane is perpendicular to the longitudinal direction of the rail.

8. Actually, the effective surface of the rail is usually very smooth, and the reflection and its influence is very critical, which may be fatal. So, the impact should be carefully considered.

Therefore, there is not enough novelty and significant to recommend it for publication in this journal.

Author Response

(The authors gave the same response as above.)

Round 2

Reviewer 3 Report

The authors have added a lot of experiments and answered most of the questions. However, some issues should be explained in detail.

1.    In Section 2 Fig.3 (b), the authors showed how the no-coplanar of laser would affect the measurement results. To verify these results, two sets of measurement equipment, (each set includes one laser plane and one camera) are required to take part in the measurement. But in Fig.14, the worn part of the rail seems to be able to be measured by one set of measurement equipment. If the experiment results in Fig.19 were obtained by one set of equipment, it can not verify that “the proposed method can improve the accuracy of rail profile measurement (Line 456)” but can only verify “the importance of the laser plane perpendicular to the longitudinal direction of rail (# 7 in last review)”. So, the experiment scene and more details about this experiment should be added.

2.    In video 1, the sentence “laser plane distance: 32.06 mm” was never changed. Is it a display error?

3.    The word “Determine” in line 129 should be lowercase. It seems a grammatical mistake in this sentence.

4.    In line 349, the camera resolution loses the unit ‘pixel’.
